# NEURAL CIRCUIT ARCHITECTURAL PRIORS FOR QUADRUPED LOCOMOTION

## ABSTRACT

Learning-based approaches to quadruped locomotion commonly adopt generic policy architectures like fully connected MLPs. As such architectures contain few inductive biases, it is common in practice to incorporate priors in the form of rewards, training curricula, imitation data, or trajectory generators. In nature, animals are born with priors in the form of their nervous system's architecture, which has been shaped by evolution to confer innate ability and efficient learning. For instance, a horse can walk within hours of birth and can quickly improve with practice. Such architectural priors can also be useful in ANN architectures for AI. In this work, we explore the advantages of a biologically inspired ANN architecture for quadruped locomotion based on neural circuits in the limbs and spinal cord of mammals. Our architecture achieves good initial performance and comparable final performance to MLPs, while using less data and orders of magnitude fewer parameters. Our architecture also exhibits better generalization to task variations, even admitting deployment on a physical robot without standard sim-to-real methods. This work shows that neural circuits can provide valuable architectural priors for locomotion and encourages future work in other sensorimotor skills.

## 1 INTRODUCTION

Learning-based approaches to quadruped locomotion commonly adopt generic policy architectures like fully connected multilayered perceptrons (MLPs) (Rudin et al., 2022; Smith et al., 2022; Agarwal et al., 2022). As such architectures contain few inductive biases, they must rely on training to develop desired behaviors. Simple objectives often fail to elicit naturalistic or robust behavior (Heess et al., 2017), so it is common in practice to incorporate priors in the form of rewards (Rudin et al., 2022), training curricula (Agarwal et al., 2022; Rudin et al., 2022), imitation data (Bin Peng et al., 2020; Merel et al., 2019b), or trajectory generators (Schaal, 2006; Iscen et al., 2019).

In nature, animals are born with priors in the form of their nervous system's architecture, which has been shaped by evolution to confer innate ability and efficient learning (Zador, 2019; Cisek, 2019). For instance, a horse can walk within hours of birth and can quickly improve with practice, and humans have strong inductive biases for perceiving and interacting with the world (Lake et al., 2017). These inductive biases are a reflection of highly structured neural circuit connectivity (Luo, 2021), which combines innate and learning mechanisms in stark contrast to generic ANN architectures.

Can such architectural priors also be useful in ANN architectures? Bhattasali et al. (2022) investigated this by introducing Neural Circuit Architectural Priors (NCAP). Using a case study of the nematode *C. elegans*, the proposed Swimmer NCAP translated neural circuits for swimming into an ANN architecture controlling a simulated agent from an AI benchmark (Tassa et al., 2020). Swimmer NCAP achieved good performance, data efficiency, and parameter efficiency compared to MLPs, and its modularity facilitated interpretation and transfer to new body designs. As such, Swimmer NCAP demonstrated several possible advantages of biologically inspired architectural priors.

However, it remained unknown whether the approach could apply to more complex animals and tasks. *C. elegans* has a nervous system of only 302 neurons and highly stereotyped connectivity, and its connectome has long been mapped (White et al., 1986). In contrast, mammalian nervous systems have millions or billions of neurons (Herculano-Houzel et al., 2006; 2007) with more variable connectivity and no mapped connectome, so it was not obvious how such circuits could inspire ANNs.

In this work, we introduce Quadruped NCAP, a biologically inspired ANN architecture for quadruped locomotion based on neural circuits in the limbs and spinal cord of mammals. Our key insights for adapting the NCAP approach are: modeling at the level of genetically defined neural populations (Danner et al., 2017; Ausborn et al., 2019; Kim et al., 2022) rather than at the level of single neurons, leveraging machine learning to compensate for gaps in detailed circuit knowledge, and introducing methodological innovations to improve the architecture's expressivity and trainability (Section 2). Together, these insights enable the architecture to successfully control quadruped locomotion, which is a much harder task than planar swimming due to its higher dimensionality and inherent instability.

We test the value of our architectural prior by comparing to an architecture without priors: the MLP, which is commonly used in robotics. Quadruped NCAP achieves good initial (untrained) performance and comparable final (asymptotic) performance to MLPs, while using less data and orders of magnitude fewer parameters. Our architecture also exhibits better generalization to task variations, even admitting deployment on a physical robot without standard domain randomization methods that are often needed for sim-to-real generalization. This work shows that neural circuits can provide valuable architectural priors for locomotion in more complex animals and encourages future work in yet more complex sensorimotor skills.

The key contributions of this work are:

1. **First genetically defined neural circuit model for quadruped robot locomotion.** We design an architecture that applies NCAP to quadruped locomotion. While many related works have been inspired by biology (Section 2), to the best of our knowledge this work is the first to use genetically defined neural circuits to control locomotion in a standard quadruped robot.

2. **Extensive evaluation in simulation.** We extensively evaluate NCAP in terms of performance, data efficiency, parameter efficiency, and generalization to terrain and body variations. NCAP learns more naturalistic gaits, with up to millions of fewer timesteps and orders of magnitude fewer parameters than MLP, while being more robust to unseen conditions.

3. **Deployment on the physical robot.** We deploy NCAP to the physical robot to test its generalization across a large sim-to-real domain gap. While MLP falls immediately due to its erratic and unstable actions, NCAP manages to walk successfully.

Our open-source code and videos are available at: https://ncap-quadruped-anon.github.io/

## 2 RELATED WORK

Our work builds on an extensive literature in neuroscience, robotics, and artificial neural networks. For conciseness, we highlight the most relevant ones to our work:

**Central Pattern Generators**   Central pattern generators (CPGs) in biology are neural circuits that produce rhythmic activity in the absence of rhythmic inputs, and they underlie movements including chewing, breathing, and locomotion. Roboticists have developed CPG-like controllers for a variety of tasks (Ijspeert, 2008; Yu et al., 2014), which can use diverse mechanisms: directly parameterizing the desired movement trajectories or footfall patterns (Iscen et al., 2019), learning to control an action space of abstract oscillators (Bellegarda & Ijspeert, 2022; Shafiee et al., 2023), designing rules-based transitions between stance/swing states (Ekeberg & Pearson, 2005), or modeling a reduced neural circuit (for instance, with 1 neuron per limb) (Lodi et al., 2018). In this work, we take a highly biologically constrained approach, designing an architecture based on cell-type-specific neural connectivity and intrinsic dynamics based on experimental data.

**Neuromechanical Models**   Neuromechanical models are used in computational neuroscience to develop insights about the interactions between the musculoskeletal system and the nervous system (Ausborn et al., 2021; Markin et al., 2016). Recently, several neuromechanical models have been developed for the rodent (Merel et al., 2019a; Tata Ramalingasetty et al., 2021) and the fly (Lobato-Rios et al., 2022; Wang-Chen et al., 2023), which will enable new understanding about how animals perform movement. In this work, we build upon insights gleaned from neuromechanical models, but our goal is not to control a realistic musculoskeletal simulation. Rather, we aim to translate insights from biology to AI and robotics, which leads us to model at a higher level of abstraction.

**Architectural Priors** Architectural priors incorporate structure into an ANN to improve performance and efficiency. For example, convolutional neural networks inspired by the visual system incorporate translation invariance over images (Lindsay, 2021). In locomotion, various kinds of architectural prior have been explored, including priors on task/body symmetries (Mittal et al., 2024; Ding & Gan, 2024), spring-loaded inverted pendulum dynamics (Ordonez-Apraez et al., 2022), modularity (Huang et al., 2020; Chiappa et al., 2022), and hierarchy (Heess et al., 2016; 2017). In this work, we explore priors based on genetically defined neural circuits, similar to connnectome-constrained networks explored in the vision literature (Lappalainen et al., 2024). Our prior is encoded through sparse and structured connectivity, weight sign constraints, weight initializations, bilateral symmetry, and intrinsic neural dynamics.

**Swimmer NCAP** In this work, we build on Swimmer NCAP (Bhattasali et al., 2022) by scaling it to more complex settings. The animals we study are more complex: mammals have much larger and less understood nervous systems than nematodes. The AI tasks we tackle are also much harder: the Quadruped is a higher-dimensional inherently unstable body, while the Swimmer is a lower-dimensional inherently stable (planar) body. We also evaluate on a physical robot, in contrast to previous work in simulation only. Thus, our work addresses problems that are especially relevant for AI. Scaling to more complex settings requires novel methodological innovations: (1) We adopt a continuous-time formulation of ANNs, in contrast to the discrete-time Swimmer. (2) We build a recurrent architecture in the Rhythm Generation (RG) module, in contrast to the feedforward Swimmer. (3) We develop a closed-loop Oscillator unit that admits period modulation, phase shifts, and entrainment, in contrast to the open-loop Oscillator unit in Swimmer that does not respond to input signals. (4) We model circuits at the level of neural populations, rather than at the level of single neurons. (5) We enable the architecture to learn unknown connections in the Afferent Feedback (AF) and Pattern Formation (PF) modules by leaving certain connectivity and signs unconstrained, in contrast to the Swimmer which fully constrained connectivity and signs. Together, these novel insights and methods can help the community advance biologically inspired architectural priors.

## 3 METHODS

We translate neuroscientific models of quadruped locomotion circuits into an ANN architecture for controlling a robot. In Section 3.1, we review key background about the neuroscience of locomotion. In Section 3.2, we describe the computational units that are building blocks in our Quadruped NCAP architecture. In Section 3.3, we describe the connectivity of NCAP and its interface with the robot.

### 3.1 BIOLOGICAL LOCOMOTION

Quadrupeds locomote by rhythmically flexing and extending their limbs in a coordinated gait to propel the body forward. They control their velocity by producing various gaits (such as walk, trot, gallop, and bound), and they adapt to different conditions using sensory information. Quadruped mammals (including mice, cats, dogs, and horses) exhibit considerable differences in appearance, but comparative anatomical studies have revealed a remarkable homology in body structure and neural circuitry between them, which makes sense given their shared evolutionary heritage (Grillner & El Manira, 2020). Neuroscience research across many animal systems has shed light on how locomotion is achieved through the complex interaction between the musculoskeletal system and neural circuits in the limbs, spinal cord, and higher brain regions (Grillner & El Manira, 2020). Surprisingly, neural circuits in the limbs and spinal cord are sufficient to produce locomotion, while higher brain regions are important to initiate and regulate locomotion (Rybak et al., 2015). Classic studies strikingly demonstrated evidence of such organization using decerebrate animals, in which most of the brain was severed from the spinal cord, yet the animal could still walk and even transition between gaits when tugged along a treadmill (Whelan, 1996). Recent studies have leveraged advances in experimental tools like molecular genetics to precisely map and manipulate locomotor circuits with cell-type specificity (Kiehn, 2016; Ausborn et al., 2021).

We summarize below a well-supported neuroscientific model of locomotor circuits (Figure 1), which is based on data from cats and transgenic mice, and which adopts the abstraction of genetically defined neural populations with rate-coded activity. For details, please refer to the referenced works.

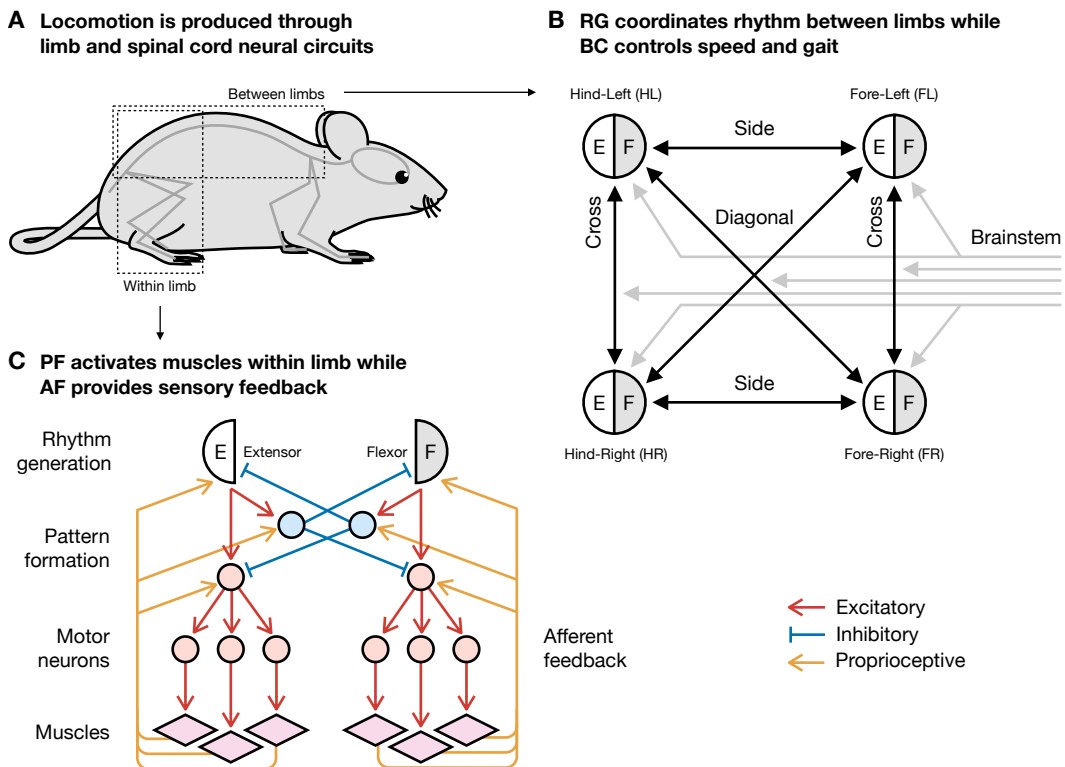

**A** Locomotion is produced through limb and spinal cord neural circuits

Between limbs

Within limb

**B** RG coordinates rhythm between limbs while BC controls speed and gait

Hind-Left (HL) Fore-Left (FL)

Side

Cross Diagonal Cross Brainstem

Side

Hind-Right (HR) Fore-Right (FR)

**C** PF activates muscles within limb while AF provides sensory feedback

Rhythm generation

Pattern formation

Motor neurons

Muscles

Extensor Flexor

Afferent feedback

← Excitatory
├ Inhibitory
← Proprioceptive

Figure 1 | **Biological Locomotion. A**, Quadruped mammals share a homologous organization of their musculoskeletal systems and neural circuits. Surprisingly, neural circuits in the limbs and spinal cord are sufficient to produce locomotion, while higher brain regions are important to initiate and regulate locomotion. **B**, Neural circuits for rhythm generation (RG) and brainstem command (BC), adapted from Danner et al. (2017). Each limb is controlled by flexor (F) and extensor (E) half-centers. Between limbs, half-centers communicate through connections that promote synchonization or alternation. Brainstem command signals modulate the half-centers and connection activations. **C**, Neural circuits for pattern formation (PF) and afferent feedback (AF), adapted from Kim et al. (2022). Within limbs, interneurons and motorneurons convert half-center states into specific muscle commands, while sensory feedback modulates the circuit at multiple levels.

**Rhythm Generation** The RG neural circuit within the spinal cord coordinates limbs to produce gait rhythms (Figure 1B). Each limb is controlled by a half-center microcircuit consisting of a flexor center with intrinsically bursting neurons and an extensor center with tonic firing neurons. The paired centers inhibit each other, leading to oscillating flexion and extension in each limb. The half-centers for the four limbs communicate through "cross" connections (cervical/lumbar commissural interneurons), "side" connections (homolateral long propriospinal neurons), and "diagonal" connections (diagonal long propriospinal neurons), thus enabling bilateral and ascending/descending communication. In essence, excitatory connections between half-centers promote synchronization and inhibitory connections promote alternation, so the speed-dependent activation of these various connections change the timing between limbs and therefore the gait. (Danner et al., 2017)

**Brainstem Command** The BC neural circuit in the brainstem conveys command signals to adjust locomotor speed and gait (Figure 1B). It does so via two pathways: one controlling speed by modulating the intrinsic period of RG oscillators, and one controlling gait by modulating the "cross" and "diagonal" RG connections that promote synchronization or alternation. (Ausborn et al., 2019)

**Pattern Formation** The PF neural circuit within each limb converts half-center states into muscle activations (Figure 1C). This circuit of interneurons and motorneurons forms a 2-level hierarchy in

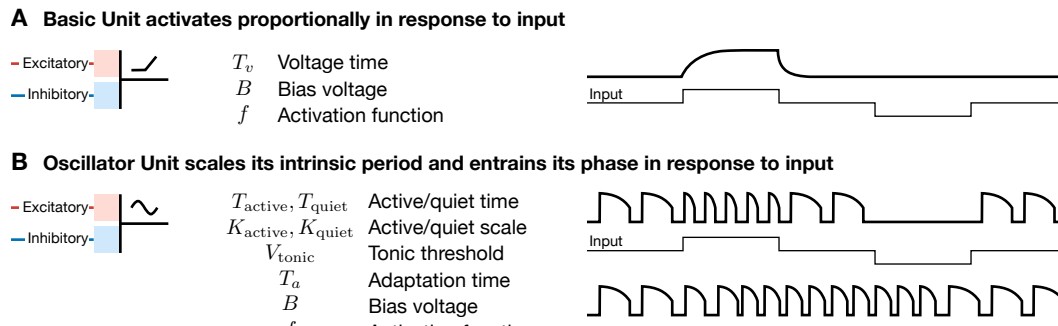

**A** **Basic Unit activates proportionally in response to input**

- Excitatory -
- Inhibitory -

$T_v$ Voltage time
$B$ Bias voltage
$f$ Activation function

Input

**B** **Oscillator Unit scales its intrinsic period and entrains its phase in response to input**

- Excitatory -
- Inhibitory -

$T_{\text{active}}, T_{\text{quiet}}$ Active/quiet time
$K_{\text{active}}, K_{\text{quiet}}$ Active/quiet scale
$V_{\text{tonic}}$ Tonic threshold
$T_a$ Adaptation time
$B$ Bias voltage
$f$ Activation function

Input

Input

Figure 2 | **Architecture Units.** The architecture uses 2 neuron types, both of which output rate-coded activity and can receive excitatory or inhibitory synaptic input. For each type, we show the circuit schematic symbol (left), key hyperparameters (middle), and an example waveform response to input (right). **A**, The Basic unit is a typical neuron that activates proportionally in response to input once its internal voltage exceeds a threshold. This unit is used for most neurons in the architecture. **B**, The Oscillator unit is a special neuron that exhibits intrinsically bursting activity in the absence of inputs. It scales its intrinsic period in response to constant input, and it entrains its phase in response to periodic input. This unit is used for the flexor half-centers in the RG module.

which flexion and extension signals are expanded in dimensionality to produce precise activation signals for each muscle. (Kim et al., 2022)

**Afferent Feedback**    The AF neural circuit within each limb uses sensory information to modulate RG and PF activity (Figure 1C). Muscle sensors produce length-, velocity-, and force-related signals, and foot sensors detect tactile stimulation. These signals trigger reflexes in PF and advance/delay half-center states in RG to entrain neural activity with musculoskeletal conditions. (Kim et al., 2022)

### 3.2 ARCHITECTURE UNITS

Our NCAP architecture adopts a continuous-time framework for modeling neurons. We find that 2 computational units can capture the cell types in this circuit: a Basic unit and an Oscillator unit (Figure 2). Many neuromechanical modeling works, including Danner et al. (2017), use biophysical neuron models that incorporate the conductances, reversal potentials, and activation/inactivation dynamics of ion channel currents (for instance, a persistent sodium current for bursting). Such complexity is not needed for AI purposes, so we follow Bhattasali et al. (2022) by simplifying these neurons to create computational units with fewer and more interpretable hyperparameters. We describe below the main properties of these units. For details and equations, please see Appendix A.1.

**Basic Unit**    This neuron model is standard in computational neuroscience. Rate-coded inputs raise or lower the internal voltage, which is leaky. If the internal voltage rises beyond a threshold, the neuron generates rate-coded output activity according to an activation function.

**Oscillator Unit**    This neuron model abstracts an intrinsically bursting neuron (Danner et al., 2017). It generates oscillating output activity in the absence of inputs. In response to inputs, it scales its active and quiet phases, and it transitions to a silent mode under strong inhibition and to a tonic mode under strong excitation. These properties enable the unit to shift its oscillation phase to pulse waveforms, and entrain its oscillation phase to periodic waveforms. We provide a detailed derivation and evaluation of this model in a concurrent manuscript (Anonymous, 2024, in submission).

### 3.3 ARCHITECTURE STRUCTURE

**Robot Body**    We target a standard robotic body in order to investigate the effectiveness of architectural priors in AI settings (Figure 3A). Unlike animals, the robot does not use highly redundant muscles that produce linear force; instead, it uses a single rotational motor per joint that produces

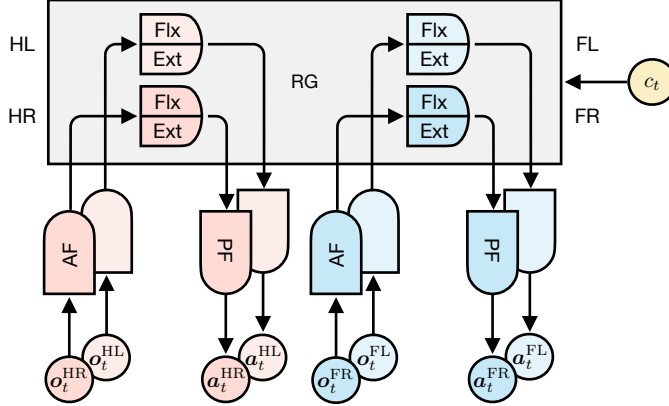

**A  Quadruped robot body**

Observations:    Joint positions
                 Joint velocities
                 Joint torques
                 Foot contact pressures

Actions:         Joint position targets

**B  Quadruped NCAP architecture**

Figure 3 | **Architecture Structure. A**, The Quadruped robot has 4 limbs, each with 3 joints (hip, thigh, calf) and 1 foot pad. The ANN agent receives proprioceptive/pressure observations, and it produces joint position target actions that are converted to actuator torque commands by a low-level PD controller. **B**, The Quadruped NCAP architecture mirrors mammalian locomotion circuits. The RG module receives a brainstem command $c_t$ that sets the speed and gait. Each limb has an AF module that uses limb observations $\boldsymbol{o}_t$ to modulate RG half-centers, as well as a PF module that converts RG half-center outputs into limb actions $\boldsymbol{a}_t$.

torque. Our architecture controlling this body must translate between biological neural circuitry and the artificial agent interface. For observations $\boldsymbol{o}_t = [\boldsymbol{q}, \dot{\boldsymbol{q}}, \boldsymbol{\tau}, \boldsymbol{f}]$, it receives joint positions $\boldsymbol{q} \in \mathbb{R}^{4 \times 3}$, joint velocities $\dot{\boldsymbol{q}} \in \mathbb{R}^{4 \times 3}$, joint torques $\boldsymbol{\tau} \in \mathbb{R}^{4 \times 3}$, and foot contact pressures $\boldsymbol{f} \in \mathbb{R}^{4 \times 1}$. For actions $\boldsymbol{a}_t = [\boldsymbol{q}_{\text{target}}]$, it produces target joint positions $\boldsymbol{q}_{\text{target}} \in \mathbb{R}^{4 \times 3}$, which are converted to actuator torque commands by a low-level PD controller (Appendix B.1). While this interface is simplified compared to biology, it reasonably approximates how muscles have a net effect of setting a joint's equilibrium position and stiffness (Shadmehr, 1993).

**Architecture Equations**   We design the Quadruped NCAP architecture to mirror mammalian locomotion circuits (Figure 3B). It is composed of several modules that together perform sensorimotor transformations within limbs and coordination between limbs:

$$(x_t^{\text{flx},l}, x_t^{\text{ext},l}) = \text{AfferentFeedback}_{\boldsymbol{\theta}}^l(\boldsymbol{o}_t^l)$$

$$(\boldsymbol{y}_t, \boldsymbol{h}_{t+1}) = \text{RhythmGeneration}_{\boldsymbol{\phi}}(\boldsymbol{x}_t, \boldsymbol{h}_t, c_t)$$

$$\boldsymbol{a}_t^l = \text{PatternFormation}_{\boldsymbol{\theta}}^l(y_t^{\text{flx},l}, y_t^{\text{ext},l})$$

with limbs $l \in \{\text{HR}, \text{HL}, \text{FR}, \text{FL}\}$, afferent inputs $\boldsymbol{x}_t \in \mathbb{R}^{4 \times 2}$ and activity outputs $\boldsymbol{y}_t \in [0, 1]^{4 \times 2}$ of RG half-centers, brainstem command $c_t \in [0, 1]$, and hidden states $\boldsymbol{h}_t$ of the continuous-time computational units. The modules are parameterized by trainable $\boldsymbol{\theta}$ and frozen $\boldsymbol{\phi}$ weights. The following subsections provide further details for each module.

**Rhythm Generation**   We design the RG module by adapting the circuit from Danner et al. (2017) to use our simplified Basic and Oscillator units. We eliminate redundant connections and introduce new nomenclature for the neuron types. As discovering the RG connections is not an aim of this work, we hand-tune the RG weights to produce the appropriate gait transitions in response to brainstem input (Figure A.3), using the reported weights from Danner et al. (2017) as a starting point. We then freeze the weights during training. For a full diagram of the RG module, please see Appendix A.2.

**Brainstem Command**   We use a brainstem command $c_t$ to control the RG module based on Ausborn et al. (2019). This latent variable from higher brain regions controls speed and gait. We treat it as a frozen hyperparameter for fixed speed tasks (Appendix A.2). If not frozen, the training usually prefers a bound gait, which is fastest and maximizes reward even on slow speed tasks. The architectural prior thus enables fine-grained gait control without additional reward/imitation priors.

**Pattern Formation**    We use a linear PF layer to map flexor/extensor half-center outputs to limb actions $a_t^l$. We initialize PF weights with coarse magnitudes and correct signs (producing negative joint positions for flexion and positive joint positions for extension) that are constrained after each update. We share weights among forelimbs and hindlimbs to exploit bilateral symmetry (Figure D.3), and we apply the overparameterization trick (Appendix A.3) to expand the dimensionality of weights during training and collapse it during testing.

**Afferent Feedback**    We use a linear AF layer to map limb observations $o_t^l$ to flexor/extensor half-center inputs. We normalize observations to the range $[-1, 1]$, then rectify them into positive and negative components, since firing rates cannot be negative. We initialize certain AF weights with coarse magnitudes to encode the known effects of limb loading and position on half-centers, and we constrain their signs (Figure D.3). We initialize unknown AF weights at zero and leave them unconstrained. We also utilize bilateral sharing and the overparameterization trick.

## 4    EXPERIMENTS

**Tasks**    We train our architecture on simulated tasks built atop the MuJoCo physics engine (Todorov et al., 2012) to control the Unitree A1 robot (Appendix B.1). The tasks are structured as 15-second episodes during which the robot must locomote forwards at a fixed target speed of Walk (0.5 m/s) or Run (1.0 m/s), and across terrains of Flat or Bumpy (Appendix B.2). The tasks provide at each timestep a reward proportional to the running speed, with maximum reward of 1 at the target speed in the forward direction (Appendix B.3). This task structure and reward design is based directly on Smith et al. (2022) in order to facilitate comparison to existing work.

**Baselines**    We compare against multilayered perceptrons (MLPs) of 2 hidden layers. By default, we compare to MLP(256,256), which is a reasonably sized architecture commonly used in the AI and robotics literature. Importantly, we choose to baseline against MLPs as they exemplify an architecture *without priors*, which contrasts with our NCAP architecture *with priors*, facilitating a clean comparison. Our goal in this work is not to compare how different classes of prior stack up generally, but rather to explore the value of neural circuit-inspired architectural priors in particular. As architectural priors are somewhat orthogonal to other forms of priors (including reward, training curricula, and imitation priors), future work could combinatorially combine our prior with others.

**Algorithm**    We train both NCAP and MLP architectures using evolution strategies (ES) to maximize episodic return (Appendix C.1). Such gradient-free optimization is easiest to use with our NCAP architecture, and it has successfully and popularly been used to train MLPs in continuous control (Salimans et al., 2017). In preliminary experiments, we compare evolution strategies to standard on-policy and off-policy reinforcement learning algorithms (Appendix C.2), and we confirm similar performance across algorithms when training MLPs on our tasks (Appendix D.1).

### 4.1    PERFORMANCE AND DATA EFFICIENCY

NCAP successfully learns to locomote across various speeds and terrains (Figure 4A). For representative examples of NCAP's behavior and neural activity, please see Videos 1.

How does NCAP compare to MLP on performance and data efficiency? NCAP achieves comparable asymptotic performance to MLP across tasks (Figure 4A). Moreover, due to its priors, an untrained NCAP achieves significantly better initial performance than an untrained MLP. The performance of NCAP improves with training as the AF and PF weights are refined. In addition, NCAP's training trajectories are less variable than MLP's.

Interestingly, NCAP appears to train more data efficiently than MLP for the harder Bumpy tasks. For instance, on the Bumpy/Run task, NCAP reaches asymptotic performance about 500 epochs (or 8 million timesteps) before MLP. However, NCAP reaches slightly lower asymptotic performance than MLP for the easier Flat tasks. We attribute this to a regularization effect in NCAP, as it is constrained in the solutions it can learn. In contrast, MLP can learn to exploit the simulator for additional performance gains, which is easier to do on Flat than Bumpy tasks.

**A** NCAP achieves comparable performance and better data efficiency

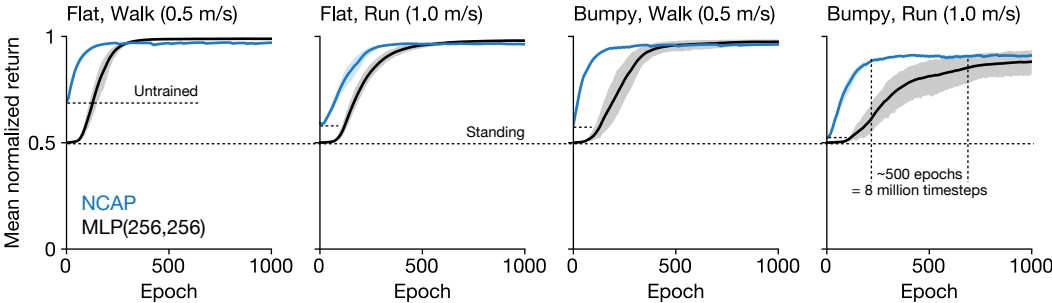

**B** NCAP exhibits more naturalistic and consistent gait patterns

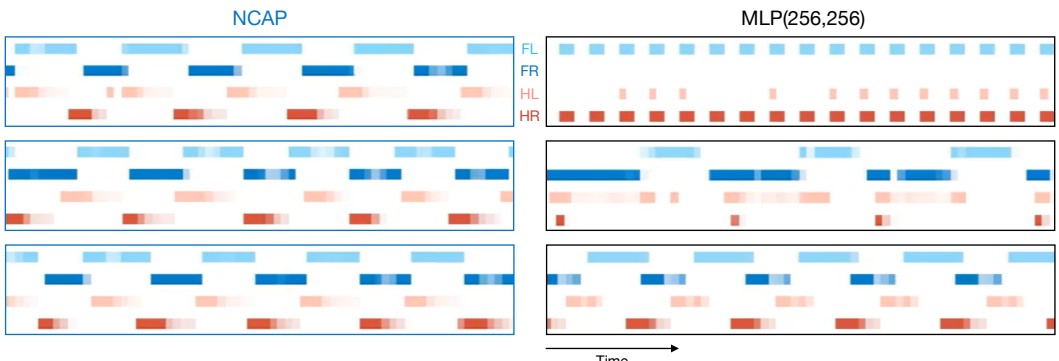

Figure 4 | **Performance and Data Efficiency. A**, Performance curves across tasks. Solid lines are mean normalized returns across 10 training seeds, each tested for 5 episodes per epoch. Shaded areas are 95% bootstrapped confidence intervals. Maximum normalized episodic return is 1, and a policy that outputs all zeros achieves 0.5 by standing still. NCAP matches or exceeds the asymptotic performance of MLP, with superior data efficiency. NCAP also demonstrates better initial performance since it is an effective prior. **B**, Footfall plots across 3 training seeds on the Flat/Walk task. Colored segments encode the foot contact pressures during stance, while blank segments indicate the limb is in swing. NCAP exhibits qualitatively more naturalistic and consistent gaits than MLP, despite their quantitatively similar asymptotic performances.

This is supported by the qualitative performance of NCAP and MLP. Using footfall plots, we examine learned gaits on the Flat/Walk task for different training seeds (Figure 4B, Videos 2). MLP develops a good walking gait on seed 3, a mediocre limping gait on seed 2, and a failed on-the-floor shuffle on seed 1; this behavior is starkly evident in Videos 2. Notably, this high variability is occluded in the performance curve (Figure 4A). In contrast, NCAP exhibits more naturalistic and consistent gaits due to its RG prior. Such gaits would require additional reward/imitation priors to elicit from MLP (Bin Peng et al., 2020; Margolis & Agrawal, 2022).

## 4.2 PARAMETER EFFICIENCY

Are the performance and data efficiency advantages of NCAP merely due to having fewer parameters? We test MLP with fewer parameters by varying the hidden layer sizes from 4 to 256. Surprisingly, performance and data efficiency degrade significantly (Figure 5A), showing that it is not merely having fewer parameters that is beneficial. Rather, the specific structure of NCAP matters.

It is this structure that enables NCAP to perform well yet require dramatically fewer parameters than MLPs (Figure 5B). During testing, NCAP uses only 92 parameters (Figure D.4), which is 3 orders of magnitude fewer than MLP(256,256) with 79,372 parameters (Figure D.6), and it is fewer than even MLP(4,4). During training, NCAP with the overparameterization trick has 708 parameters (Figure D.5), which is 2 orders of magnitude smaller than MLP(256,256), and it is comparable to MLP(16,16). Notably, many of these MLPs are much smaller than the sizes typically used in practice.

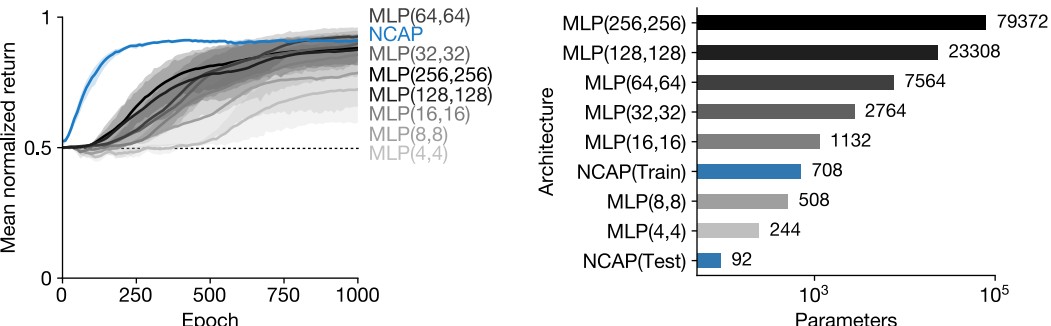

Figure 5 | **Parameter Efficiency. A**, Performance curves across MLP sizes on the Bumpy/Run task. Smaller MLPs achieve lower asymptotic performance and worse data efficiency. Therefore, having fewer parameters is insufficient to account for NCAP's advantages. **B**, Parameter count across architectures (log scale). NCAP(Test) has fewer parameters than MLP(4,4), and even NCAP(Train) with the overparameterization trick (Appendix A.3) has orders of magnitude fewer parameters than MLPs at typical sizes.

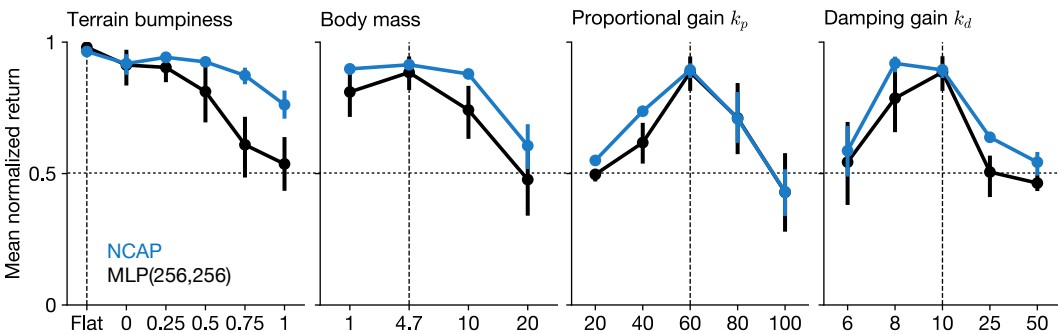

Figure 6 | **Generalization to Terrain and Body Variations. A**, Performance on unseen terrain and body variations. Architectures are trained in the condition indicated with a vertical dashed line, then tested in altered conditions. The variations included terrain bumpiness (parameterized from 0 to 1; Figure B.1), body mass (in kilograms), or proportional/damping gains of the low-level PD controller. NCAP's generalization matches, and often exceeds, that of MLP.

Moreover, in settings like actor-critic reinforcement learning that often use similarly sized actor and critic networks as well as moving average copies of those networks, the number of required parameters can quadruple. NCAP's parameter efficiency could be particularly advantageous for deployment in resource-constrained environments, like a robot's onboard compute.

### 4.3 GENERALIZATION TO TERRAIN AND BODY VARIATIONS

How well does NCAP generalize to unseen environments compared to MLP? We evaluate the architectures across a variety of terrain and body variations (Figure 6). In each setting, the architectures are trained in one condition, then tested in altered conditions. Across these variations, NCAP's generalization matches, and often exceeds, that of MLP. Surprisingly, MLP performance degrades dramatically on Bumpy terrain, despite the differences in bumpiness seeming minor by human standards (Figure B.1). In contrast, NCAP performs more robustly.

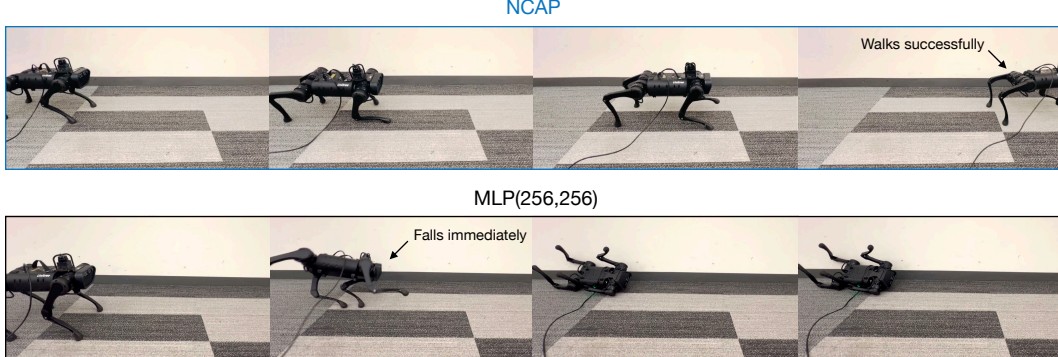

Figure 7 | **Generalization to the Physical Robot.** Video frames (Videos 3) of a representative deployment trial on the physical robot. NCAP walks successfully despite the large domain gap, while MLP falls immediately due to its erratic and unstable actions.

### 4.4 GENERALIZATION TO THE PHYSICAL ROBOT

How well does NCAP generalize to the real world compared to MLP? We deploy the architectures to a physical Unitree A1 quadruped robot after training on the Bumpy/Walk task (Appendix B.4). We expect the domain gap to be large since we do not perform controller tuning or system identification with the physical robot, the simulated training does not apply any domain randomization task priors (Peng et al., 2018), and the architectures do not have mechanisms for online adaptation.

MLP falls immediately due to its erratic and unstable actions, often launching the robot aggressively into the wall (Figure 7). Despite our best efforts, we cannot elicit a successful walking trial. In contrast, NCAP is remarkably robust to the large domain gap, walking successfully on most trials, though with less smoothness than in simulation (Figure 7). We attribute NCAP's success to a combination of the RG module maintaining a stable rhythm in the face of sensor noise and the AF module triggering corrective responses in the face of perturbations. Lastly, we deploy an untrained NCAP and observe that it is stable and produces slight walking movements with small foot displacements (Videos 3).

## 5 DISCUSSION

In this work, we introduce Quadruped NCAP, a biologically inspired ANN architecture for quadruped locomotion based on neural circuits in the limbs and spinal cord of mammals. Our architecture achieves good initial performance and comparable final performance to MLPs, while using less data and orders of magnitude fewer parameters. Our architecture also exhibits better generalization to task variations, even admitting deployment on a physical robot without standard sim-to-real methods.

**Limitations**   Our study faces several limitations. First, we rely on a hand-tuned RG module and BC command, which might not be the optimal parameters that a learning-based approach could discover. Second, we neglect musculoskeletal factors in the quadruped action space that could make learning easier or more robust.

**Future Work**   We focus on fixed speed locomotion in this work, but an obvious next step is to enable the RG to transition between gaits in a speed-dependent manner, which may largely involve a higher-level controller that alters brainstem commands for different speeds. Another extension is to add postural adjustment, turning, and righting mechanisms to the architecture based on understanding of the underlying neural circuits. Finally, as architectural priors are somewhat orthogonal to other forms of priors, it may be beneficial to train NCAP with additional reward, task, or imitation priors.

Overall, we believe that this work shows that neural circuits can provide valuable architectural priors for locomotion in more complex animals and encourages future work in yet more complex sensorimotor skills.

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

# A  ARCHITECTURE DETAILS

## A.1  ARCHITECTURE UNITS

**Basic Unit**  The Basic unit is a typical rate-coded neuron with continuous dynamics, which integrates weighted input signals to generate a target $x$ for the internal voltage $v$.

$$\text{Hyperparameters: } T_v, B, f$$

$$x = \text{clip}(B + \sum_i w_i y_i, -1, 1)$$

$$\frac{T_v}{4}\frac{dv}{dt} = x - v$$

The neuron's activation function $f$ determines the voltage-firing relationship:

$$y = f(v), \quad f = \text{clip}(v, 0, 1)$$

**Oscillator Unit**  The Oscillator unit is a relaxation oscillator with discrete-continuous dynamics on 2 variables: a discrete internal voltage $v \in -1, +1$ and a continuous adaptation variable $a \in [0, 1]$.

$$\text{Hyperparameters: } T_a, T_{\text{active}}, T_{\text{quiet}}, K_{\text{active}}, K_{\text{quiet}}, V_{\text{tonic}}, B, f$$

$$T'_{\text{active}} = \frac{4T_{\text{active}}}{T_a}, \quad T'_{\text{quiet}} = \frac{4T_{\text{quiet}}}{T_a}$$

The adaptation thresholds $a^{0,1}_{\text{quiet, active}}$ are calculated that determine the min/max values of adaptation $a$ at which the internal voltage $v$ jumps between active $(+1)$ and quiet $(-1)$ states:

$$a^0_{\text{active}} = \frac{1 - \exp(T'_{\text{quiet}})}{1 - \exp(T'_{\text{active}} + T'_{\text{quiet}})}$$

$$a^0_{\text{quiet}} = a^0_{\text{active}} \cdot \exp(T'_{\text{active}})$$

$$a^1_{\text{active}} = \frac{1 - \exp(T'_{\text{quiet}} \cdot K_{\text{quiet}})}{1 - \exp(T'_{\text{active}} \cdot K_{\text{active}} + T'_{\text{quiet}} \cdot K_{\text{quiet}})}$$

$$a^1_{\text{quiet}} = a^1_{\text{active}} \cdot \exp(T'_{\text{active}} \cdot K_{\text{active}})$$

Depending on the strength of the input signal at a given time, the quiet-to-active and active-to-quiet adaptation thresholds interpolate between the calculated min/max values. Adaptation $a$ exponentially decays towards 0 when the neuron is active (the neuron depletes the adaptation variable) and towards 1 when quiet (the neuron replenishes the adaptation variable). Voltage $v$ jumps instantaneously to a new state when an adaptation threshold is reached.

$$x = \text{clip}(B + \sum_i w_i y_i, -1, 1)$$

$$z = \text{clip}(x, 0, 1)$$

$$a_{\text{active}} = \text{interpolate}(z, a^0_{\text{active}}, a^1_{\text{active}})$$

$$a_{\text{quiet}} = \text{interpolate}(z, a^0_{\text{quiet}}, a^1_{\text{quiet}})$$

$$\frac{T_a}{4}\frac{da}{dt} = \begin{cases} 0 - a & \text{if } v = +1 \quad \text{(active)} \\ 1 - a & \text{if } v = -1 \quad \text{(quiet)} \end{cases}$$

$$v^{(t+dt)} := \begin{cases} -1 & \text{if } a^{(t)} \leq a^{(t)}_{\text{active}} \text{ and } x^{(t)} \leq V_{\text{tonic}} \quad \text{(active} \rightarrow \text{quiet)} \\ +1 & \text{if } a^{(t)} \geq a^{(t)}_{\text{quiet}} \text{ and } x^{(t)} \geq 0 \quad\quad \text{(quiet} \rightarrow \text{active)} \\ v^{(t)} & \text{otherwise} \end{cases}$$

The neuron's activation function $f$ determines the voltage-adaptation-firing relationship:

$$y = f(v, a, x), \quad f = \begin{cases} \text{interpolate}(a, 0.5, 1.0) & \text{if } v = +1 \quad \text{(active)} \\ 0 & \text{if } v = -1 \quad \text{(quiet)} \end{cases}$$

We provide a detailed derivation and evaluation of this model in a concurrent manuscript (Anonymous, 2024, in submission).

## A.2 RHYTHM GENERATION (RG) MODULE

The RG circuit diagram can be visualized in full (Figure A.1) or through a progressive breakdown (Figure A.2). The RG weights are tuned to produce the appropriate gait transitions in response to increasing brainstem command (Figure A.3).

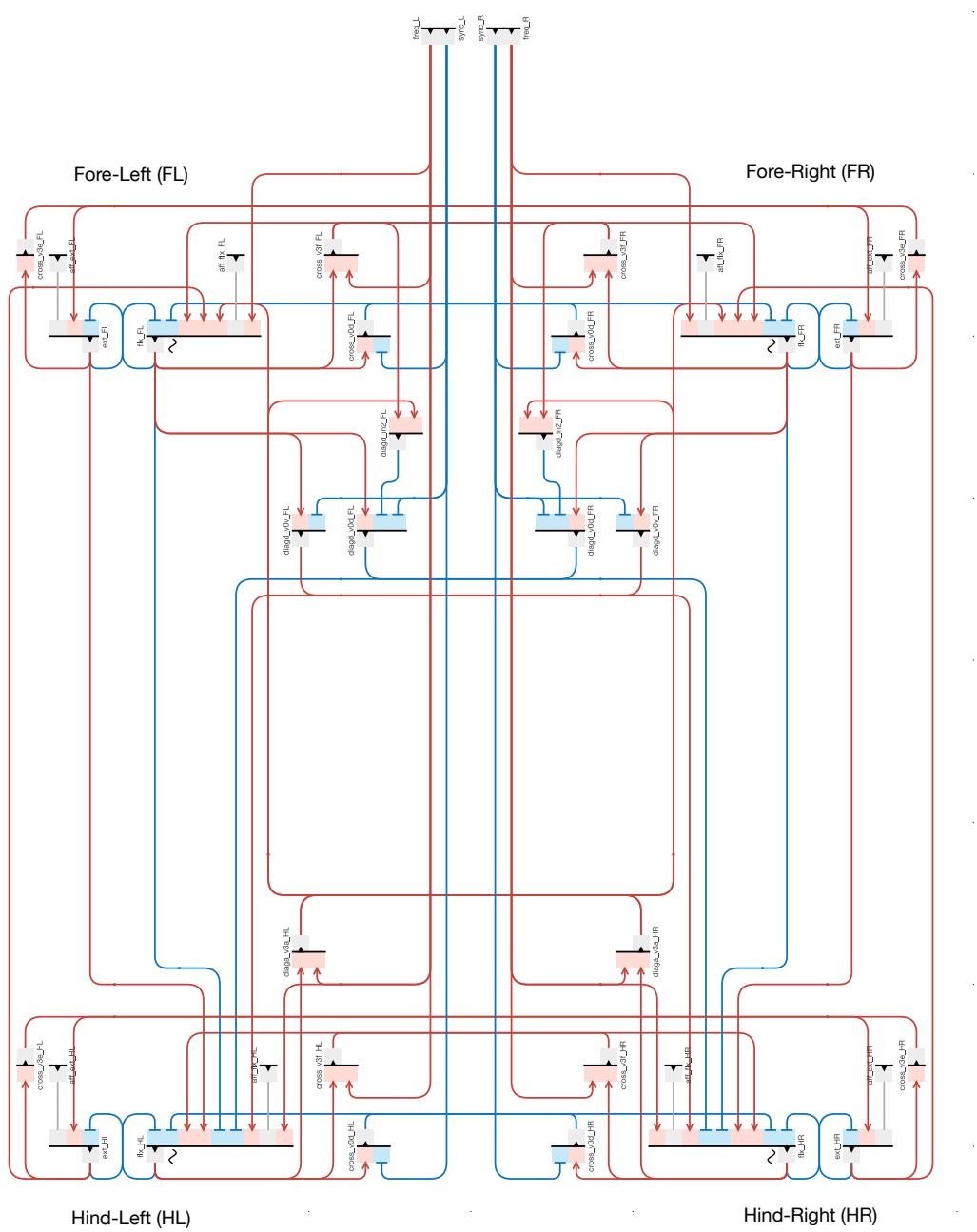

Figure A.1 | **Rhythm Generation, Full.** Neural circuits for rhythm generation (RG) and brainstem command (BC) at cell-type-specific resolution, adapted from Danner et al. (2017). For a progressive breakdown, please see Figure A.2.

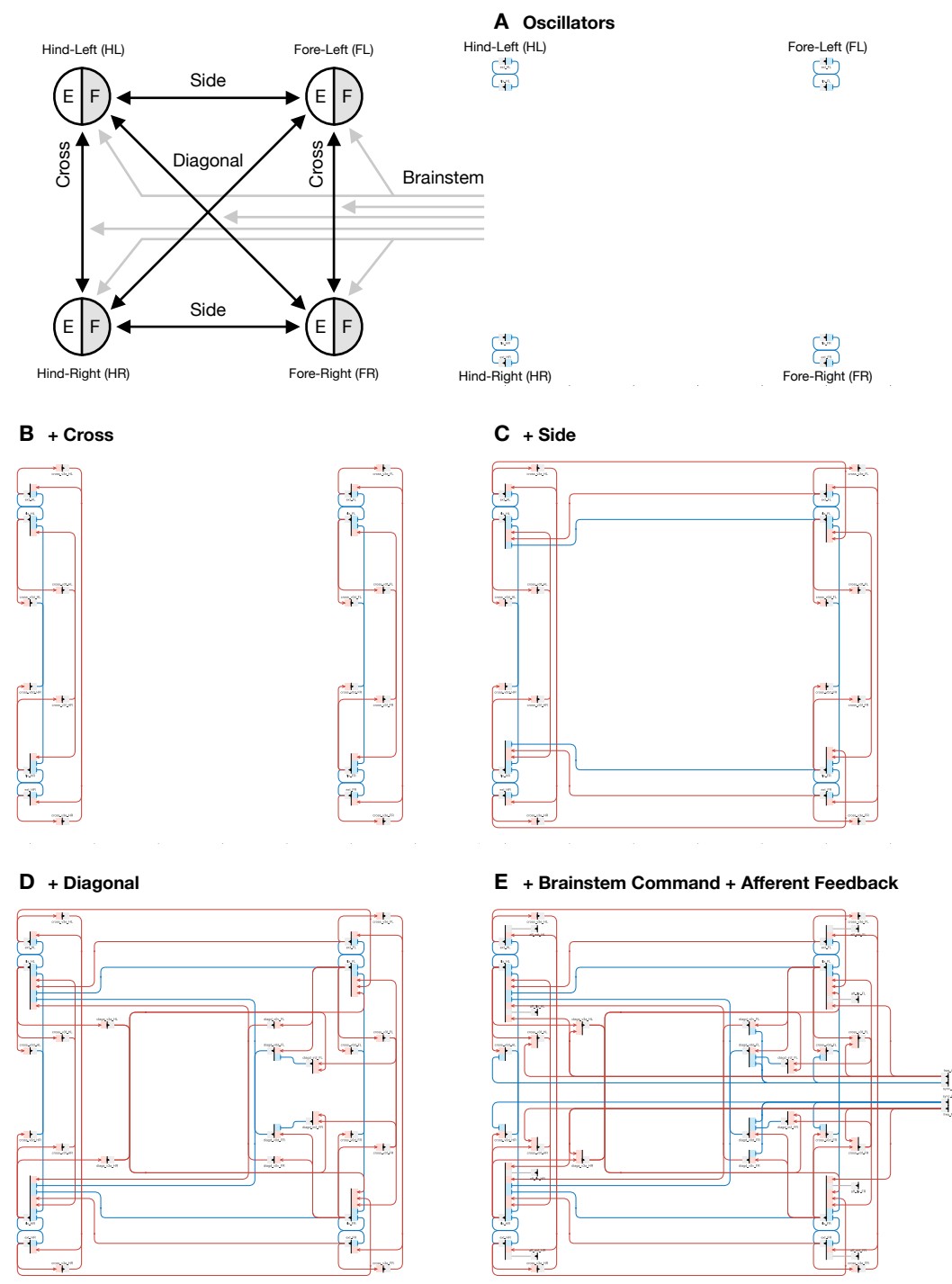

Figure A.2 | **Rhythm Generation, Breakdown.** Neural circuits for rhythm generation (RG) and brainstem command (BC) at cell-type-specific resolution, adapted from Danner et al. (2017). Each limb is controlled by **(A)** reciprocally inhibiting half-centers consisting of a flexor Oscillator unit and an extensor Basic unit. Between limbs, half-centers communicate through **(B)** cross, **(C)** side, and **(D)** diagonal connections, and they are modulated by **(E)** brainstem command and afferent feedback.

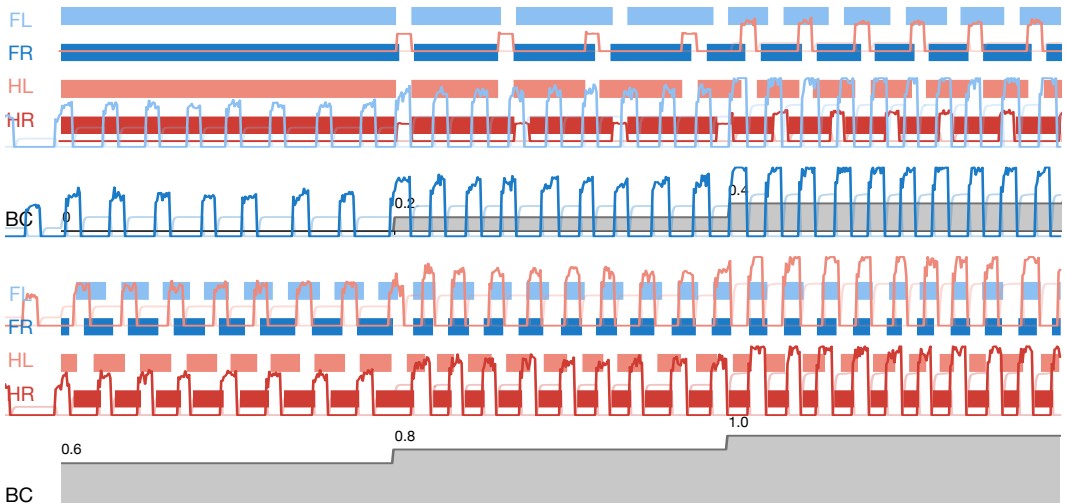

Figure A.3 | **Rhythm Generation, Gait Transitions.** Predicted footfall plots from RG extensor half-center state. The RG circuit transitions between gaits (stand → walk → trot → bound) as the brainstem command (BC) increases from 0 to 1 (shown in 5-second increments). This enables a higher-level controller to modulate the gait for variable speeds.

## A.3  OVERPARAMETERIZATION TRICK

We discover an important technique to improve the trainability of our architecture, which we call "the overparameterization trick".

Consider a linear layer $y = Wx$. During training, the weight matrix $W$ can be artificially expanded in dimensionality along the row dimension and/or column dimension to produce a larger weight matrix $W'$. The former produces a correspondingly larger output vector $y'$, while the latter necessitates a correspondingly larger input vector $x'$. The larger output $y'$ can be transformed into the original output $y$ by summing expanded elements. The original input $x$ can be transformed into the larger input $x'$ by copying and concatenating elements. Thus, the expanded layer maintains the original input and output dimensions, but with a larger weight matrix and new hardcoded expansion and compression operations at the interface. We find empirically that such overparameterization improved learning, presumably due to special dynamics of traversing high-dimensional loss landscapes with our point-based evolutionary strategies algorithm (Appendix C.1), but better theoretical understanding of this phenomenon is needed.

During testing, the overparameterized weight matrix $W'$ can be collapsed by summing along the expanded rows and/or columns to produce a smaller matrix with the size of the original $W$. Mathematically, this does not change the computation, as it exploits the linearity of matrix multiplication.

For a simple example, consider the case of 1D inputs and outputs.

If the weight is expanded row-wise:

$$y = wx \quad \underset{\text{expand}}{\longrightarrow} \quad y = \sum_{\text{rows}} \begin{bmatrix} w'_1 \\ w'_2 \end{bmatrix} x = (w'_1 + w'_2)x \quad \underset{\text{compress}}{\longrightarrow} \quad y = wx$$

If the weight is expanded column-wise:

$$y = wx \quad \underset{\text{expand}}{\longrightarrow} \quad y = \begin{bmatrix} w'_1 & w'_2 \end{bmatrix} \begin{bmatrix} x \\ x \end{bmatrix} = (w'_1 + w'_2)x \quad \underset{\text{compress}}{\longrightarrow} \quad y = wx$$

For this work, the technique enables NCAP to leverage the benefits of training with overparameterized networks using an expanded architecture (Figure D.5), which is compressed for testing (Figure D.4).

## B  ENVIRONMENT DETAILS

### B.1  SIMULATED ROBOT

We train our architecture on simulated tasks built using DeepMind Composer (Tassa et al., 2020) atop the MuJoCo physics engine (Todorov et al., 2012) using a Unitree A1 robot model imported from MuJoCo Menagerie (Zakka et al., 2022).

The observation and action interfaces with the agent are normalized to $[-1, 1]$. The action space is designed with a default standing pose corresponding to actions of 0 and joint limits corresponding to actions of $\pm 1$.

A proportional-derivative (PD) controller is employed to convert the target joint positions $\boldsymbol{q}_{\text{target}}$ generated by the agent into joint torque commands for the actuators:

$$\boldsymbol{\tau}_{\text{target}} = k_p(\boldsymbol{q}_{\text{target}} - \boldsymbol{q}) - k_d \cdot \dot{\boldsymbol{q}}$$

with joint positions $\boldsymbol{q}$, joint velocities $\dot{\boldsymbol{q}}$, proportional gain $k_p$, and derivative gain $k_d$.

The simulation runs with a control timestep of 0.03 seconds and a physics timestep of 0.001 seconds.

### B.2  TASK STRUCTURE

The tasks are formulated as 15-second episodes. The agent incurs a penalty and the episode resets if the robot falls over or if any part of its base touches the ground. At the start of each episode, the friction coefficient of the foot is randomized. The agent is trained to move at a fixed velocity of Walk (0.5 m/s) or Run (1.0 m/s) over either Flat or Bumpy terrain (Figure B.1).

### B.3  TASK REWARDS

We adopt the reward function from Smith et al. (2022), which uses the forward linear velocity $v_x$ in the robot frame, the pitch angle $\theta_y$ in the world frame, the target forward velocity $v_{\text{target}}$ in the world frame, and the angular yaw velocity $\omega_z$ in the robot frame. The overall reward combines a linear velocity reward and an angular velocity penalty. The former encourages the agent to maintain a forward velocity parallel to the ground within a margin of the target velocity, while the latter reduces unnecessary rotations to encourage stable and straight motion:

$$
\begin{aligned}
r &= r_{\text{linear}} + r_{\text{angular}} \\
v &= v_x \cos(\theta_y) \\
r_{\text{linear}} &= \begin{cases} 1 & v \in [v_{\text{target}}, 2v_{\text{target}}] \\ 0 & v \in (-\infty, -v_{\text{target}}] \cup [4v_{\text{target}}, \infty) \\ 1 - \frac{|v - v_{\text{target}}|}{2v_{\text{target}}} & \text{otherwise} \end{cases} \\
r_{\text{angular}} &= -0.1|\omega_z^2|
\end{aligned}
$$

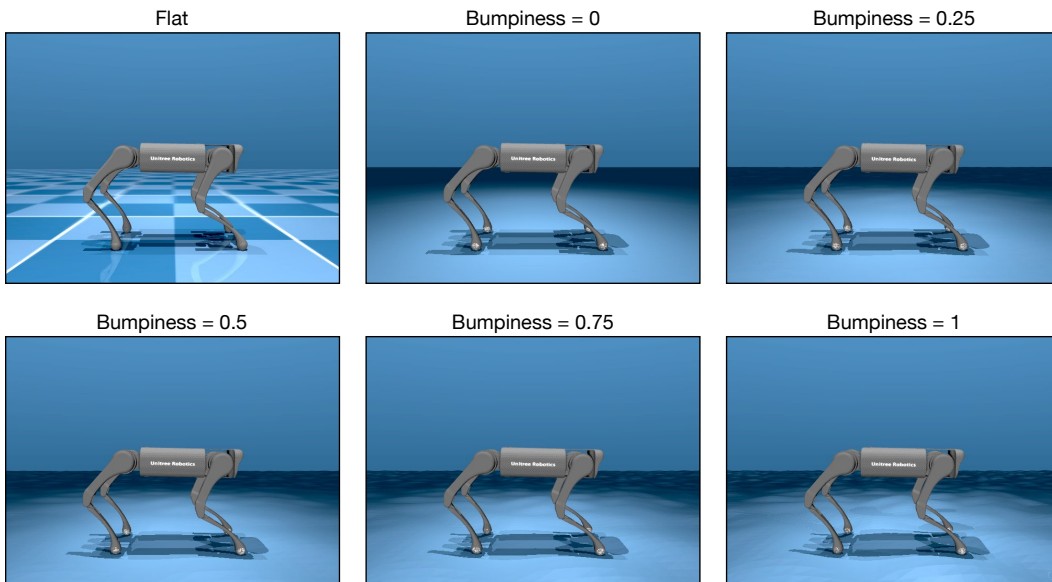

Figure B.1 | **Terrain Bumpiness.** The Flat and Bumpy terrains have different frictions. The Bumpy terrain allows bumpiness to be parameterized between 0 and 1.

### B.4 PHYSICAL ROBOT

**Deployment Setup**    The experiments are conducted on the Unitree A1 quadruped robot with 12 degrees of freedom (3 per leg). The robot has sensors for joint positions, joint torques, joint velocities, and foot contact pressures, and it has actuators that produce commanded torques at each joint. The communication between the robot and the control policy is performed through Lightweight Communications and Marshalling (LCM) (Huang et al., 2010) and a pybind build of `unitree_legged_sdk`. The control policy is deployed on the robot from a local workstation through a LAN connection, in order to ensure low-latency and high-reliability communication.

**Experimental Protocol**    The deployment process includes an initial calibration phase to verify joint offsets and zero torque sensors, ensuring accurate sensor readings and actuator control. Notably, no system identification is performed with our simulated robot model. Despite substantial efforts to tune the MLP policy, including adjustments to proportional/derivative gains and initial placement, it struggles to generalize effectively to the physical robot. In contrast, our NCAP policy exhibits surprising robustness.

# C  Algorithm Details

## C.1  Evolution Strategies

**Augmented Random Search (ARS)**  ARS is an evolutionary strategies algorithm that spawns offspring networks by randomly perturbing the parent network's weights, scores offspring according to a fitness function, and selects the next parent by taking a fitness-weighted average of offspring weights. It can be approximately viewed as estimating gradients of the fitness function. (Mania et al., 2018)

| Hyperparameter | ARS |
| --- | --- |
| Population size | 32 |
| Mutation scale | 0.1 |
| Number of workers | 32 |

Table 1 | **Hyperparameters for ES algorithm.** Further details in our provided code.

## C.2  Reinforcement Learning

**Proximal Policy Optimization (PPO)**  PPO is an on-policy reinforcement learning algorithm known for its stability. By maintaining a proximity constraint between the new and old policies, it balances exploration and exploitation, making it a reliable choice for continuous control tasks. (Schulman et al., 2017)

**Distributed Distributional Deep Deterministic Policy Gradient (D4PG)**  D4PG is an off-policy reinforcement learning algorithm known for its data efficiency. By incorporating parallel rollouts and a distributional value function, it efficiently reuses gathered experience to learn continuous control tasks. (Barth-Maron et al., 2018)

| Hyperparameter | D4PG | PPO |
| --- | --- | --- |
| Actor network layers | (256, 256) | (64, 64) |
| Actor network activation function | ReLU | Tanh |
| Critic network layers | (256, 256) | (64, 64) |
| Critic network activation function | ReLU | Tanh |
| Observation normalizer | Mean-Std | Mean-Std |

Table 2 | **Hyperparameters for D4PG and PPO algorithms.** Further details in the algorithm implementations from Pardo (2021).

## C.3  Computational Resources

Training is performed on a high-performance computing cluster running the Linux Ubuntu operating system. The ES algorithm is parallelized over 32 cores. The RL algorithms are parallelized over 16 cores as minimal speedups are observed beyond that on our tasks; this kind of nonlinear scaling is consistent with reported performance tests in other work (Salimans et al., 2017).

# D  SUPPLEMENTAL EXPERIMENTS

## D.1  PERFORMANCE AND DATA EFFICIENCY

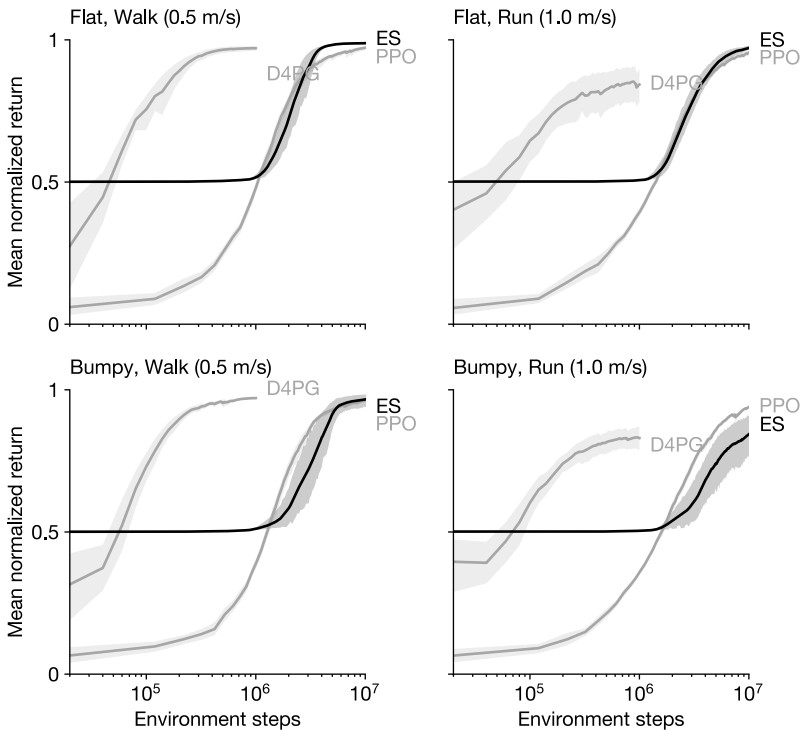

Figure D.1 | **Performance and Data Efficiency, MLP Architecture, Various Algorithms.** er-
formance curves across various tasks. Solid lines are mean normalized returns across 10 training
seeds, each tested for 5 episodes per epoch. Shaded areas are 95% bootstrapped confidence intervals.
The $x$-axis is environment steps to ensure a fair data efficiency comparison between algorithms,
although the time efficiency between algorithms is significantly different. ES achieves comparable
performance to PPO and D4PG on our tasks.

## D.2  PARAMETER EFFICIENCY

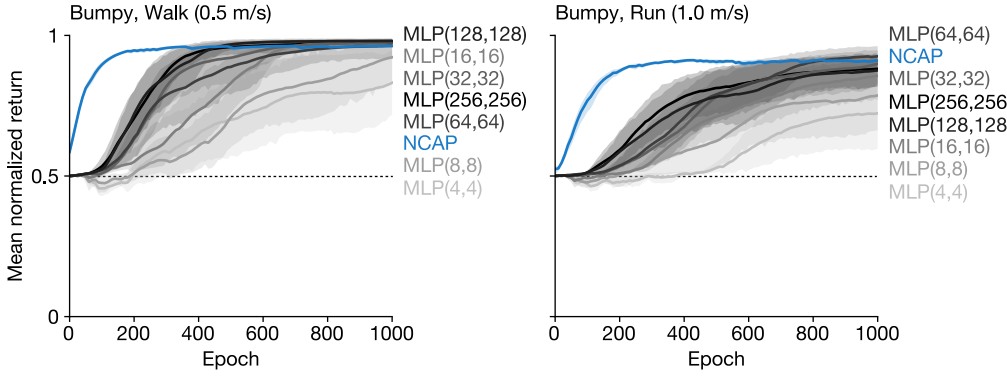

Figure D.2 | **Parameter Efficiency, NCAP/MLP Architectures, Various Tasks.** Performance
curves across MLP sizes on Bumpy tasks. Smaller MLPs achieve lower asymptotic performance and
worse data efficiency, which is more extreme in the harder Bumpy/Run task.

## D.3    INTERPRETABILITY

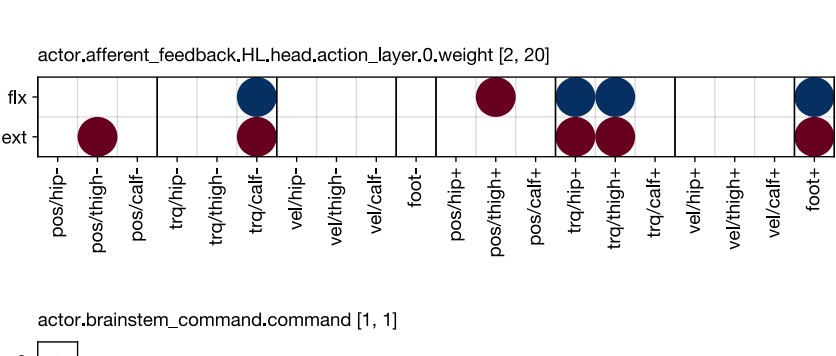

Figure D.3 | **Interpretability, NCAP Architecture, Untrained, Compact.** Weights plot of the untrained NCAP. Weight sign is encoded in color, with excitatory (positive) weights as red and inhibitory (negative) weights as blue. Weight magnitude is encoded in circle lightness and diameter. The non-zero elements of NCAP's weights are initialized with coarse magnitudes and constrained signs.

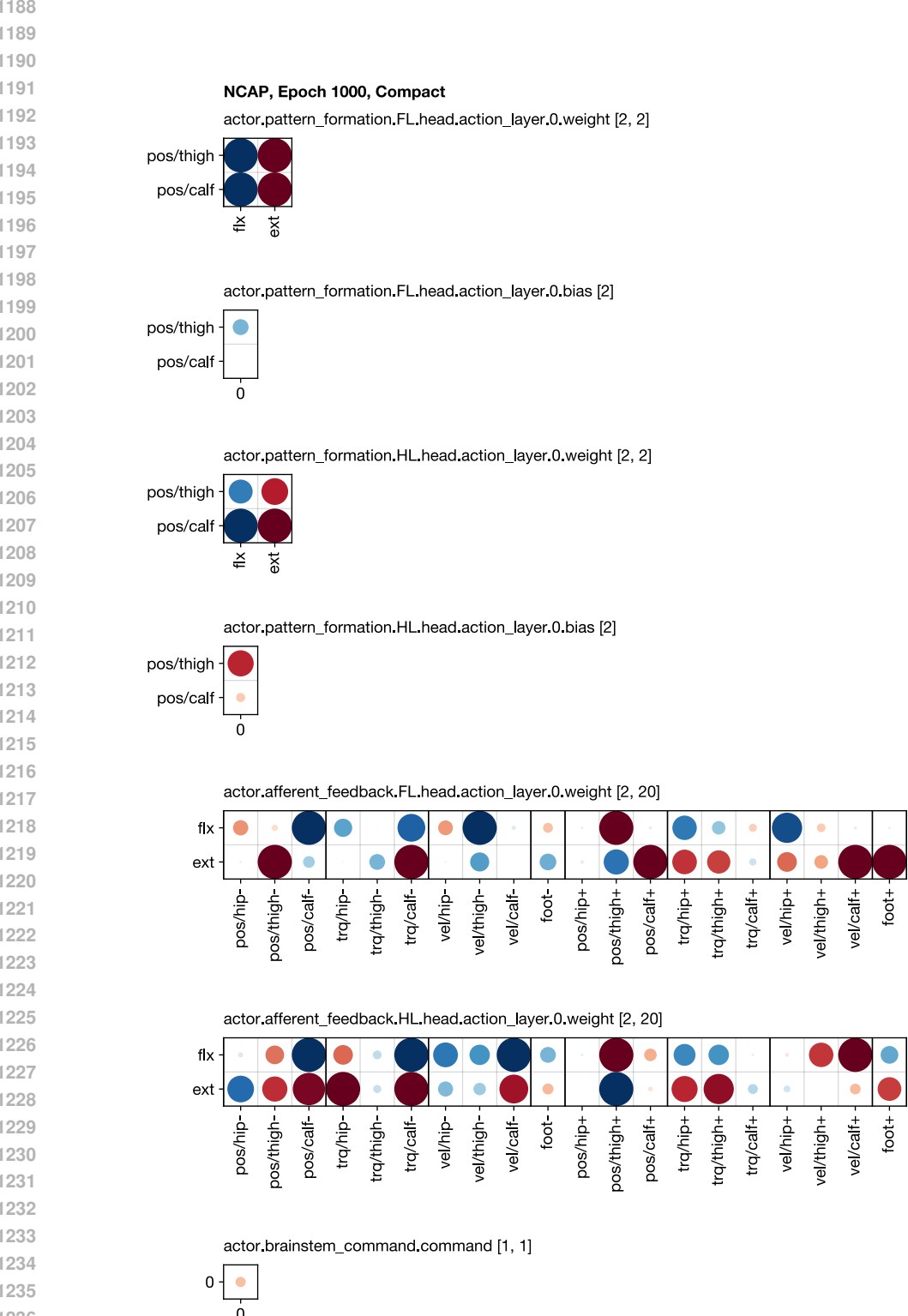

Figure D.4 | **Interpretability, NCAP Architecture, Trained, Compact.** Weights plot of the trained NCAP in its compact/testing variant.

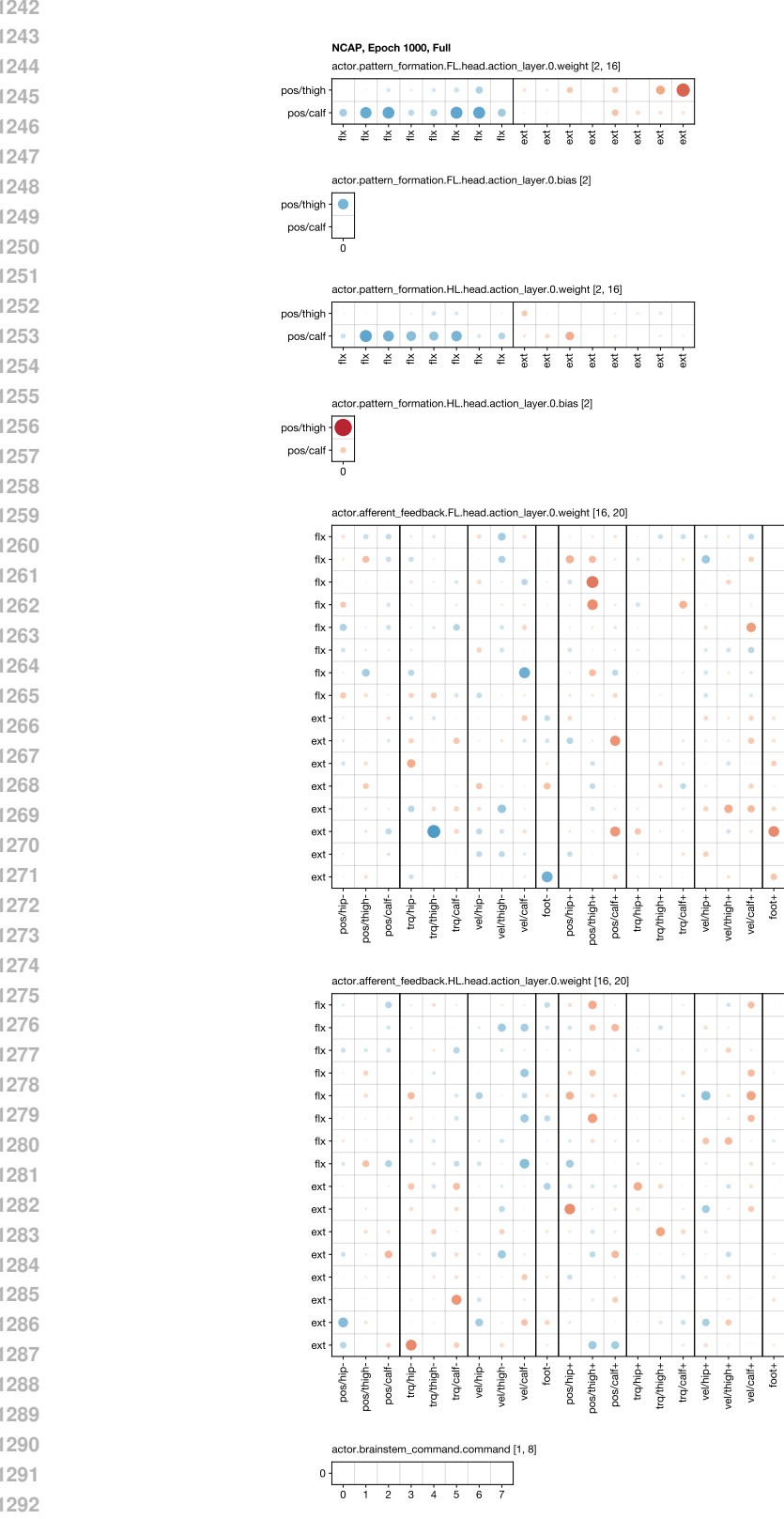

Figure D.5 | **Interpretability, NCAP Architecture, Trained, Full.** Weights plot of the trained NCAP in its overparameterized/training variant.

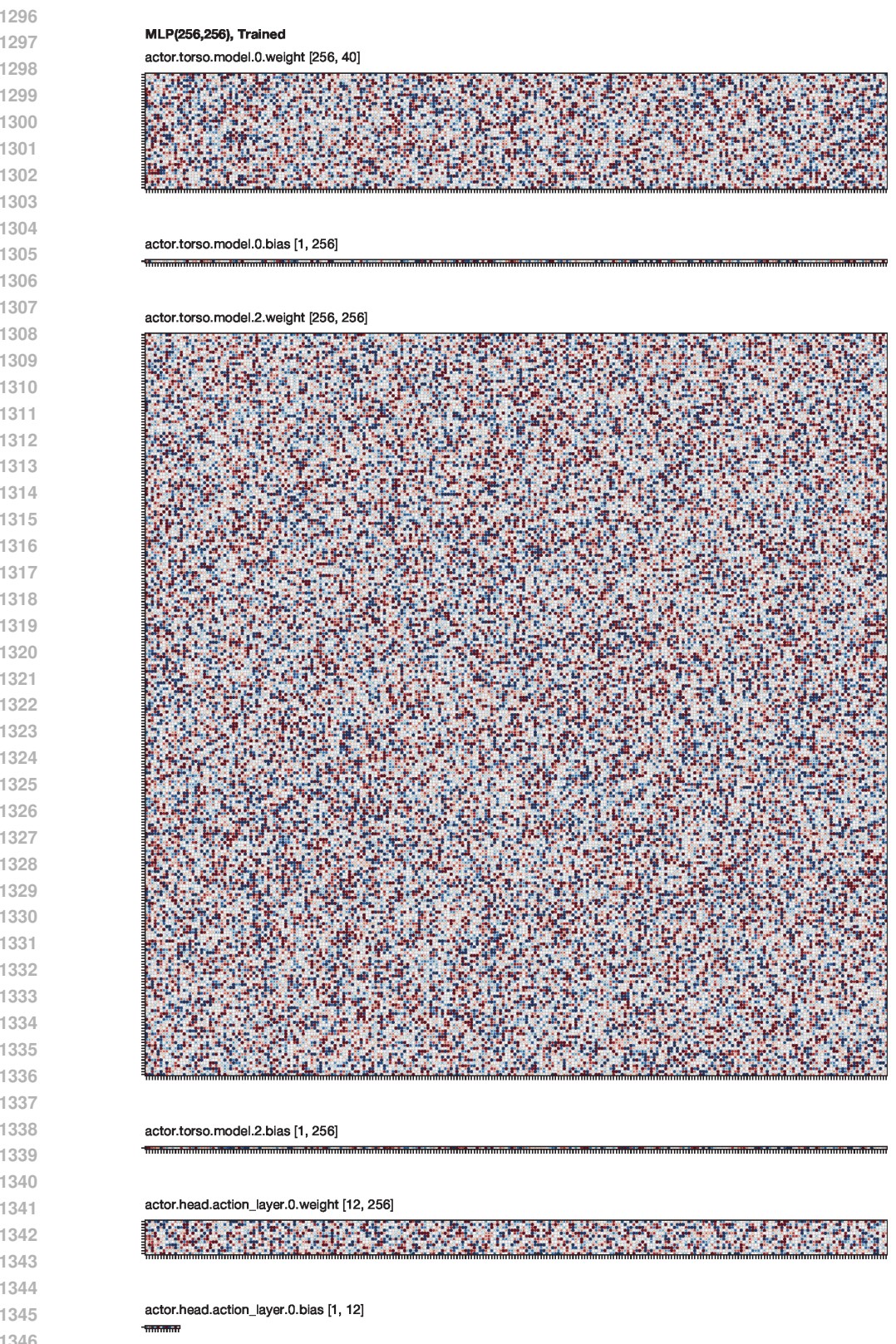

Figure D.6 | **Interpretability, MLP Architecture, Trained, Full.** Weights plot of the trained MLP. Such weights are difficult to interpret, as the rows and columns of hidden weights lack fixed meanings, and weight signs can change freely during training.

## D.4 VARIABLE SPEED

We perform additional experiments to investigate the ability of Quadruped NCAP to handle a more complex task in which the robot must transition between speeds within an episode.

**Task**  We train our architecture on a simulated task built atop the MuJoCo physics engine to control the Unitree A1 robot, as in the fixed speed tasks (Section 4). The task is structured as a 15-second episode during which the robot must locomote forwards across Flat terrain at a variable target speed randomly sampled every 5 seconds from the set $\{0.0, 0.25, 0.5, 0.75, 1.0\}$ m/s. Therefore, there are up to 3 distinct target speeds per episode to track, and the current target speed $v_{\text{target},t}$ is included within the observation at each timestep. The task provides at each timestep a sparse reward, with maximum reward of 1 at the target speed in the forward direction and with reward decaying to zero at a margin around the target speed. This reward design is based on Rudin et al. (2022).

The reward function uses the forward linear velocity $v_x$ in the robot frame, the pitch angle $\theta_y$ in the world frame, and the target forward velocity $v_{\text{target}}$ in the world frame. The overall reward provides a linear velocity reward that encourages the agent to maintain a forward velocity parallel to the ground within a margin of the target velocity:

$$r = r_{\text{linear}}$$
$$v = v_x \cos(\theta_y)$$
$$v_{\text{margin}} = 0.2 \text{ m/s}$$
$$r_{\text{linear}} = \exp\left[-4\left(\frac{v - v_{\text{target}}}{v_{\text{margin}}}\right)^2\right]$$

**Architecture**  We extend the brainstem command $c_t$ to be the output of a simple linear transform:

$$c_t = w \cdot v_{\text{target},t} + b$$

with the current target speed $v_{\text{target},t}$ from the observation, and weight $w$ and bias $b$ treated as hyperparameters. Therefore, the brainstem command can vary as the target speed changes in order to control speed and gait. From a grid search, we find that $w = 1.2$ and $b = 0$ works well. We retune RG and AF hyperparameters for this task, and we also add a new connection from BC to RG extensor centers so that both half-centers can modulate movement vigor based on speed.

**Algorithm**  We train both NCAP and MLP architectures using evolution strategies (ES), as in the fixed speed tasks (Section 4).

**Results**  NCAP successfully learns to locomote and transition between speeds. For a representative example of NCAP's behavior and neural activity, please see Videos 4.

NCAP uses the target speed to modulate the brainstem command that controls the RG circuit. This enables it to alter its gait in a speed-dependent manner, as predicted from RG activity (Figure A.3).

MLP also learns to alter its speed, but its posture and gait are not naturalistic without additional priors (Videos 4). In particular, at a target speed of 0 m/s, MLP learned different ways of not moving forwards, but there were often artifacts like shaky limbs. It is typical to use reward/task priors (e.g. on joint motions or body height) to attempt to minimize such artifacts, but even state-of-the-art methods with such priors report that artifacts are visible (Rudin et al., 2022).

