# OpenReview forum: "Neural Circuit Architectural Priors for Quadruped Locomotion"
_ICLR.cc/2025/Conference — Submitted to ICLR 2025_

### Official Review · Reviewer_Sb2E · 2024-10-31

**Soundness:** 3
**Presentation:** 3
**Contribution:** 3
**Rating:** 6
**Confidence:** 3

**Summary:**

The paper presents a biologically inspired neural architecture for quadruped locomotion, named Quadruped NCAP, designed to replicate mammalian neural circuits in the limbs and spinal cord. In contrast to traditional artificial neural networks (ANNs) like multilayer perceptrons (MLPs), which are widely used in robotic locomotion but lack inductive biases, Quadruped NCAP incorporates neural architectural priors, allowing it to operate with better data efficiency and computational resource savings. The study demonstrates that this architecture achieves comparable performance to MLPs while requiring fewer parameters and displaying enhanced generalization to varied terrains and task settings.

**Strengths:**

Biologically-Informed Structural Priors: Integrates neural circuit patterns, allowing the model to leverage structural priors that support data-efficient learning and adaptability across various terrains.

Comparison with Traditional ANNs: Contrasts NCAP with conventional artificial neural networks like multilayer perceptrons (MLPs), highlighting that NCAP requires fewer parameters and less data, avoiding the need for large datasets and extensive computational resources.

Real-World Testing and Robust Generalization: Evaluates NCAP’s performance in simulations and on a physical robot, demonstrating effective generalization and adaptability without additional domain adaptation.

**Weaknesses:**

Hand-Tuned Parameters: The model’s Rhythm Generation (RG) module and Brainstem Command are set manually rather than learned, which could make it harder to adapt to new tasks or find the best settings automatically.

Fixed Speed Limitation: The model currently supports only one set speed, meaning it cannot adjust its walking or running pace, which limits flexibility in different environments.

Limited to Locomotion Tasks: This study only tests NCAP on walking tasks, so it’s unclear how well this approach would work for other movements or types of tasks.

**Questions:**

How might the model’s learning stability or convergence be affected if RG and Brainstem parameters were learned rather than hand-tuned? Would automatic learning potentially introduce instability or require a specialized training regime?

If parameters were learned for one locomotion task, could they be transferred or fine-tuned for different tasks (e.g., varied speeds, uneven terrain)? How transferable are these parameters across different conditions?
Does the current architecture allow for speed and gait adaptation on the fly, and if not, what modifications would be necessary to support dynamic speed changes in real time?

Would automatically learning these parameters add significant computational overhead, and if so, how could this be mitigated? Are there any techniques that could allow for efficient online learning or adaptation of these parameters?

Could the authors compare Quadruped NCAP with other models that incorporate different types of architectural priors (e.g., symmetry, task-based priors) to assess specific advantages of neural circuit-based priors?

---

### Official Review · Reviewer_LDgZ · 2024-11-01

**Soundness:** 2
**Presentation:** 3
**Contribution:** 2
**Rating:** 5
**Confidence:** 3

**Summary:**

This work is inspired by the four limb and spinal nerve circuits involved in mammal, and explores the advantages of ANN architecture in quadrupedal locomotion control. Compared to traditional neural networks such as MLP, neural circuits can provide valuable prior information for locomotion and achieve good performance with minimal data and parameters.

**Strengths:**

1.Inspired by the ability of horses to stand and walk within a few hours of birth, a biologically inspired ANN architecture based on limbs and spinal cord neural circuits provides a novel perspective in the field of quadrupedal locomotion control.
2.Compared to traditional neural network architectures such as MLP, the model parameters of this research method are very small, and the simulation experiment results have achieved similar performance.
3.It has shown generalization performance in different tasks, even in situations that have not been encountered during training, such as different terrains and speeds, this research method can maintain relatively good performance.

**Weaknesses:**

1.The author mainly introduces the various components of the model in the method section, and the roles played by these components during the training and inference processes are not clearly explained.
2.Normally, using MLP as the neural network and online reinforcement learning algorithm PPO can control quadruped robots to walk on more complex terrains such as stairs. However, this research method was only tested on flat and bump terrains, and did not verify whether quadruped NAVP can achieve motion control of quadruped robots in more complex scenarios.
3. It has been demonstrated in the experiment that the prior knowledge of quadrupled NAVP is valuable. However, the PPO trained solely on speed tracking as a reward signal does not perform well and cannot prove its performance advantage compared to reinforcement learning.

**Questions:**

1. The article mentions manually adjusting the RG module and BC command, but what is the process of adjustment? What role did each module play during the training and deployment process?
2. The locomotion control of quadruped robots on multiple terrains is already quite mature, why not train models on more complex terrains? Is it because the generalization of the method is limited?
3. The online reinforcement learning algorithm PPO is relatively easy to implement for motion control of robots on flat ground and can be deployed to physical objects, but using only speed tracking rewards cannot train useful PPO algorithms. Although your experiment has proven the value of prior knowledge, what are the performance advantages of your method?Are there any issues with the subjects of the comparative experiment?

---

### Official Review · Reviewer_sjvx · 2024-11-03

**Soundness:** 3
**Presentation:** 3
**Contribution:** 3
**Rating:** 6
**Confidence:** 4

**Summary:**

This research introduces "Quadruped NCAP," a new way to make four-legged robots walk by copying how animal nervous systems work. Instead of using traditional artificial intelligence that needs lots of training data, this system copies the natural nerve circuits found in animal legs and spinal cords, using only 92 parameters (while traditional methods use nearly 80,000) yet works remarkably well. The system helps robots walk naturally and adapt to different surfaces, demonstrating that copying designs from nature can create better and more efficient robot control systems.

**Strengths:**

**Originality & Quality**
- First successful translation of mammalian neural circuits to robot control
- Dramatically efficient: uses only 92 parameters vs 79,372 in traditional methods
- Thoroughly tested in both simulation and real world, with open-source code

**Clarity & Significance**
- Clear explanation and visualization of complex biological-to-AI translation
- Major practical impact: simpler system that works better in real world
- Opens new direction for bio-inspired AI, showing that copying nature's designs can make robot control more efficient and effective

**Weaknesses:**

1. The theoretical contribution of NN structure is weak, lacking formal analysis of why this biological architecture works better than traditional ones.
2. The tasks (basic locomotion) are relatively simple compared to SOTA robotics challenges, making the parameter efficiency less impressive.
3. Unclear how well NCAP generalizes from simple to complex tasks, with limited analysis of the adaptation process.
4. Missing comparisons with other bio-inspired approaches like CPG and SNN.
5. No justification for why the chosen biological components are optimal for robot control versus other possible biological mechanisms.
6. Hand-tuned RG module could be hiding complexity that's just shifted from parameters to manual engineering.
7. Performance evaluation focuses on basic metrics (walking success).
8. No ablation studies showing which biological components are truly necessary for performance gains.
9. Missing analysis of computational cost and real-time performance requirements versus traditional approaches.
10. Limited exploration of how this approach could scale to more complex behaviors beyond basic locomotion.

**Questions:**

1. While the paper shows successful quadruped deployment, the novelty is questionable as the core neural architecture is heavily based on Swimmer NCAP, with limited theoretical justification for the adaptations made from simple to complex systems.
2. The paper lacks comparisons with other bio-inspired approaches (especially CPG-based methods which are standard in locomotion control).

---

### Official Review · Reviewer_QxDf · 2024-11-04

**Soundness:** 3
**Presentation:** 3
**Contribution:** 2
**Rating:** 6
**Confidence:** 4

**Summary:**

This paper presented a model for learning motion control of a quadruped robot. The model is bioinspired by incorporating a CPG and firing rate neurons. Experiments show the model can control a robot walking on flat ground well and the model has a higher parameter efficiency than typical MLP models.

**Strengths:**

The model is a firing rate model modelled with differential equations, which suggests the model has internal dynamics. It could be the reason why it is more parameter-efficient than MLP on continuous control tasks.  The experiments are not only conducted on a simulated robot but also on a real robot.

**Weaknesses:**

While this work is interesting, some of the contributions are overstated. The authors should be careful to claim the "First neural circuit model for quadrupedal robot locomotion", and investigate deeper into older papers from 10, 20, or even 30 years ago, there were papers that used complex CPGs to generate gaits for legged robot locomotion.

Besides, the key contributions 2 and 3 stated on page 2 are not contributions but simply good practices when researching bioinspired models for robots. The texts from lines 124 to 132 read like contributions but need a better summary.

There is no comparison between the model with SOTA RL models.

The model is not presented very well mathematically. The corresponding section should be in the main text but not the appendix.

**Questions:**

1. How was the model trained? It is important but not clearly stated.
2. What is exactly the MLP for? There are many RL methods with MLP. The baseline is too vague.
3. Does the robot use any sensor except the joint sensors? Can it only walk straight? Can it change gaits?
4. What does the Brainsem command refer to? How does this part of the model work? Are there any reference papers for it? Are there any plots or examples for the command?

The score for the paper could either be up or down according to the responses.

---

### Meta-Review · Area_Chair_HFei · 2024-12-26

**Metareview:**

The authors provided a summary of contributions and reviewers' concerns.
The authors raised concerns about reviewers R3 and R4. It might be more convincing to have another round of reviews with updated reviewers.

As mentioned by the authors, this paper is about a neural circuit model for the neuroscientific study of motor control, target to ICLR’s “applications to neuroscience” track.
The authors presented MuJoCo simulation experiments to control the Unitree A1 robot without neuroscientific experiments.
R3 asked for baselines against SOTA robotics methods, which didn't satisfy the authors with a focus on applications to neuroscience.
The AC didn't find any neuroscientific experimental results. It would be nice to show some (even very simple) neural recordings and behavior analysis during motor tasks, as a proof for its “applications to neuroscience”.

**Additional Comments On Reviewer Discussion:**

Reviewers R1 and R2 updated their scores to 6 after the discussion phase.
AC made the evaluation based on the final manuscript, discussions, opinions, and scores.

---

### Decision · Program_Chairs · 2025-01-22

Reject